# Perioperative Tailored Treatments for Gastric Cancer: Times Are Changing

**DOI:** 10.3390/ijms24054877

**Published:** 2023-03-02

**Authors:** Daniele Lavacchi, Sara Fancelli, Eleonora Buttitta, Gianmarco Vannini, Alessia Guidolin, Costanza Winchler, Enrico Caliman, Agnese Vannini, Elisa Giommoni, Marco Brugia, Fabio Cianchi, Serena Pillozzi, Giandomenico Roviello, Lorenzo Antonuzzo

**Affiliations:** 1Clinical Oncology Unit, Careggi University Hospital, 50134 Florence, Italy; 2Department of Experimental and Clinical Medicine, University of Florence, 50134 Florence, Italy; 3Medical Oncology Unit, Careggi University Hospital, 50134 Florence, Italy; 4Unit of Digestive Surgery, Careggi University Hospital, 50134 Florence, Italy; 5Department of Health Science, University of Florence, 50134 Florence, Italy

**Keywords:** perioperative GC, HER2, MSI-H, Claudin18.2

## Abstract

Resectable gastric or gastroesophageal (G/GEJ) cancer is a heterogeneous disease with no defined molecularly based treatment strategy. Unfortunately, nearly half of patients experience disease recurrence despite standard treatments (neoadjuvant and/or adjuvant chemotherapy/chemoradiotherapy and surgery). In this review, we summarize the evidence of potential tailored approaches in perioperative treatment of G/GEJ cancer, with a special focus on patients with human epidermal growth factor receptor-2(HER2)-positive and microsatellite instability-high (MSI-H) tumors. In patients with resectable MSI-H G/GEJ adenocarcinoma, the ongoing INFINITY trial introduces the concept of non-operative management for patients with complete clinical-pathological-molecular response, and this could be a novel and potential practice changing strategy. Other pathways involving vascular endothelial growth factor receptor (VEGFR), fibroblast growth factor receptor (FGFR), claudin18 isoform 2 (CLDN18.2), and DNA damage repair proteins are also described, with limited evidence until now. Although tailored therapy appears to be a promising strategy for resectable G/GEJ cancer, there are several methodological issues to address: inadequate sample size for pivotal trials, underestimation of subgroup effects, and choice of primary endpoint (tumor-centered vs. patient-centered endpoints). A better optimization of G/GEJ cancer treatment allows maximizing patient outcomes. In the perioperative phase, although caution is mandatory, times are changing and tailored strategies could introduce new treatment concepts. Overall, MSI-H G/GEJ cancer patients possess the characteristics to be the subgroup that could receive the most benefit from a tailored approach.

## 1. Introduction

Gastric cancer (GC) is the fifth most common cancer and the third cause of cancer-related mortality worldwide [1,2]. When diagnosis occurs in metastatic disease, the prognosis is poor and radical surgery is not routinely recommended [3].

In the setting of resectable disease, multimodal treatments, including perioperative chemotherapy, showed a significant survival improvement compared with surgery alone. At first, the Medical Research Council Adjuvant Gastric Infusional Chemotherapy (MAGIC) and FNCLCC/FFCD trials demonstrated the benefit of perioperative chemotherapy in survival and in recurrence-rate reduction. Patients with resectable gastric or gastro-esophageal junction (G/GEJ) adenocarcinoma were assigned to either surgery alone or perioperative chemotherapy (i.e., epirubicin, cisplatin and fluorouracil [ECF] in the MAGIC trial and cisplatin and fluorouracil in the FNCLCC/FFCD trial). Five-year survival rates were higher in the perioperative-chemotherapy group compared to the surgery group (36% vs. 23% in MAGIC, 38% vs. 24% in FNCLCC/FFCD trial) [4,5].

Although docetaxel showed a remarkable efficacy in first- and further-line treatments for metastatic G/GEJ adenocarcinoma, both alone and in combination with cisplatin and fluorouracil [6,7], toxicities associated with combination regimens were so relevant to suggest replacement of cisplatin with oxaliplatin. This new combination, which consisted of perioperative fluorouracil, leucovorin, oxaliplatin and docetaxel (FLOT), was evaluated in the phase III FLOT4-AIO trial that demonstrated the efficacy of FLOT as a perioperative therapy over ECF in G/GEJ adenocarcinoma. The median overall survival (OS) was 50 months in the FLOT arm vs. 35 months in the ECF arm, and the pathological complete regression (pCR) rate was 16% vs. 6%, respectively [8,9]. A recent Italian case series (RealFLOT study) confirmed the feasibility and safety of the FLOT regimen in a less prognostically favorable selected patient population. In this study, the pCR rate was 7.3%, highlighting the need to explore novel combination therapies in order to improve clinical outcomes and long-term efficacy of perioperative treatments [10].

In advanced G/GEJ cancer, the introduction of human epidermal growth factor receptor-2 (HER2)-directed therapies and immune checkpoint inhibitors (ICIs) in combination with chemotherapy has improved survival. Moreover, novel targets could significantly change the continuum of care [11,12,13,14,15].

In this review, we describe the current strategies to improve the clinical outcome in resectable GC patients with the addition of tailored treatments to perioperative chemotherapy.

## 2. Tailored Treatments

### 2.1. HER2-Directed Therapies

Over the past few years, increasing attention has developed around targeted therapies, and substantial changes have been made in therapeutic choices and strategies in most malignant tumors [16]. One of the first molecular targets was HER2. The development of the specific monoclonal antibody (moAb) trastuzumab, the first targeted agent for solid malignancies, led to a drastic change in the clinical course of HER2-positive breast cancer patients [17]. In GC patients, previous studies have reported that the HER2 protein expression level ranged from 7.3% to 22.1% [18]. The association between HER2 status and prognosis has been described in several studies, with a general agreement on the negative prognostic role of HER2 overexpression [19,20,21]. The randomized controlled ToGA trial showed that the addition of trastuzumab to chemotherapy offered a meaningful survival gain in advanced GC patients overexpressing HER2 (mOS 13.8 months vs. 11.1 months, HR 0.74; 95% CI 0.60–0.91; *p* = 0.0046) [13]. Kurokawa and colleagues later demonstrated that HER2 overexpression was an independent prognostic factor in patients with resectable GC (HR 1.59; 95% CI 1.24–2.02; *p* = 0.001), and they also showed that HER2 intra-tumoral heterogeneity was frequent and did not affect prognosis. In this study, 180 HER2-positive cases were identified among 1148 GC patients, assessed by immunohistochemistry (IHC) and fluorescence in situ hybridization. The trial showed that HER2-positive GC was more commonly related to intestinal-type adenocarcinoma and upper stomach location. Other factors, including the pT and pN stages, did not show any correlation with HER2 overexpression [22]. As previously described, the results from the AIO-FLOT4 trial have set a new standard of care for resectable GC patients fit for intensive treatment. However, more than half of the patients in the FLOT group experienced disease recurrence within 3 years and the estimated OS at 5 years was 45% [9] (Figure 1).

Given the results of the AIO-FLOT4 trial in the perioperative setting, the next step was to evaluate the combination of trastuzumab and FLOT as perioperative treatment in patients with HER2-positive, locally advanced esophagogastric adenocarcinomas.

HER-FLOT was a multicenter, phase II study aiming at evaluating the toxicity and activity of trastuzumab in combination with chemotherapy in the perioperative setting. This trial reached the primary endpoint (pCR > 20%) achieving a pCR rate of 21.4%. In fact, 12 out of 56 patients had no viable neoplastic cells at the time of surgery, as assessed by a central pathologist, and 14 patients (25.0%) had subtotal response (<10% residual tumor). The most frequently observed grade (G) 3–4 adverse events (AEs) were quite similar to the FLOT4 trial, and only one case of severe heart failure related to trastuzumab was reported. pCR was also a surrogate of survival outcome. Overall, median disease-free survival (DFS) was 42.5 months and the 3-year OS rate was 82.1% [23].

The PETRARCA trial explored the role of dual HER2 blockade (trastuzumab and pertuzumab) in combination with chemotherapy (FLOT) compared to chemotherapy alone in HER2-positive GC patients who were amenable to a perioperative strategy. Trastuzumab and pertuzumab were administered every 3 weeks for 3 preoperative and 3 postoperative cycles, followed by 9 maintenance cycles. The primary endpoint was the pCR rate that was achieved in 35% of patients (*n* = 14) in the experimental arm vs. 12% of patients (*n* = 5) in the control group (*p* = 0.019). Among secondary endpoints, node-negativity and radical-surgery (i.e., R0) rates were met. The 2-year OS rate was 84% in the experimental arm and 77% in the control arm. Median DFS was not reached in the experimental arm vs. 26 months in the FLOT arm (HR 0.58, 95% CI 0.28–1.19, *p* = 0.130), with 2-year DFS of 70% and 54%, respectively. The safety profile of the experimental group was characterized by a predominance of diarrhea, neutropenia and leukopenia as high G AEs [24].

The negative results from the JACOB trial, which evaluated the addition of dual anti-HER2 blockade to a taxane-free polychemotherapy regimen, led to a premature stop of the PETRARCA trial and failure to transition to a phase III study [25]. However, PETRARCA’s investigators pointed out the high pCR and pN0 rates in the experimental arm at the cost of slightly higher gastrointestinal and hematopoietic toxicity.

The ongoing, international, randomized phase II EORTC INNOVATION trial has the purpose to evaluate the addition of trastuzumab or trastuzumab plus pertuzumab to chemotherapy in the perioperative treatment of patients with HER2-positive G/GEJ adenocarcinoma. Preliminary results are expected in 2023 [26].

In support of the anti-HER2 blockade strategy, results from some retrospective and prospective experiences conducted in Asian patients have recently been published.

Data from 45 Asian patients with HER2-positive stage II-III GC were retrospectively analyzed. Twenty-nine patients received trastuzumab plus FLOT and sixteen received chemotherapy alone. The primary endpoint was the objective response rate (ORR). In the trastuzumab + FLOT arm, the ORR was 72.4% and the disease control rate (DCR) was 89.7%, while in the FLOT group the ORR and DCR were 43.8% and 87.5%, respectively. The 2-year OS rates were 78.1% and 73.9%, respectively (*p* = 0.932). Although the study did not reach statistical significance in primary and ancillary endpoints, the addition of trastuzumab provided a numerically high tumor response rate and a promising pathological regression compared to the control group: the tumor regression rate to grade Ia/Ib was obtained in 44.8% of patients [27].

The Japan Clinical Oncology Group (JCOG) conducted the multi-institutional two-arm open label randomized phase II Trigger trial to evaluate the efficacy and safety of S-1-cisplatin in combination with trastuzumab vs. S-1-cisplatin alone for patients with HER2-positive locally advanced GC. The primary endpoint was OS. The study enrolled 46 patients of the preplanned 130 due to slow accrual and was prematurely ended in 2021. The ORR was higher in the experimental group than in the control group (66.7% vs. 36.4%, *p* = 0.08), even though the difference was not statistically significant. pCR rates were 50.0% and 22.7% (*p* = 0.07), respectively, and the percentages of patients who experienced pathological downstage were 22.7% and 50.0% (*p* = 0.07), respectively [28]. Although of interest, this study did not add conclusive data on the role of anti-HER2 blockade in patients with locally advanced disease.

After several disappointing or inconclusive results about the use of anti-HER2 agents other than trastuzumab, the recent results from the phase II DESTINY-Gastric01 trial with the antibody-drug conjugate trastuzumab deruxtecan (T-DXd) renewed interest in advanced GC. The excellent performance of T-DXt in terms of ORR (51.3% vs. 14.3%) and OS (median 12.5 vs. 8.4 months; IC 95% 0.39–0.88, HR 0.59; *p* = 0.01) compared with physicians’ choice of chemotherapy has led to planning the phase II EPOC2003 study (NCT05034887) with the aim to evaluate T-DXd as a neoadjuvant treatment for patients with HER2-positive G/GEJ adenocarcinoma in six Japanese centers [29,30].

Finally, the randomized phase III RTOG 1010 study included a total of 202 patients diagnosed with locally advanced HER2-positive esophageal adenocarcinoma. Patients were randomized to receive either chemoradiotherapy (CROSS scheme) plus trastuzumab or chemoradiotherapy alone. The primary endpoint was DFS, which was not met. Median DFS was 19.6 months (95% CI 13.5–26.2) with chemoradiotherapy plus trastuzumab compared to 14.2 months (10.5–23.0) for chemoradiotherapy alone (HR 0.99, 95% CI 0.71–1.39, *p* = 0.97). In conclusion, although well tolerated, the addition of trastuzumab to the neoadjuvant trimodality treatment did not result in any benefit, neither in terms of pCR nor of DFS [31].

A summary of the main trials is reported in Table 1.

### 2.2. Immune Checkpoint Blockade

As in many solid tumors, the standard of care in G/GEJ cancer is about to change with the introduction of ICIs in combination with chemotherapy as the first line of treatment [15].

Programmed cell death ligand 1 (PD-L1) is a protein expressed in different immune system cells (e.g., T lymphocytes, epithelial cells, endothelial cells, macrophages, dendritic cells). Cancer cells also use PD-L1 to evade the anti-tumor immune response inhibiting cytotoxic T-cell activity. Moreover, neoplastic cells have the ability to upregulate anti-cytotoxic T-lymphocyte-associated antigen-4 (CTLA-4), leading to the formation of a co-inhibitory pathway to avoid host immune responses. Antibodies against the checkpoint proteins programmed death-1 (PD-1) (nivolumab, pembrolizumab), PD-L1 (atezolizumab, avelumab, durvalumab) and CTLA-4 (ipilimumab, tremelimumab) were shown to be effective in reactivating protective T-cell activity [32].

One of the most important predictors of benefit from immunotherapy is the deficient mismatch repair/microsatellite instability-high (dMMR/MSI-H) status, leading to the agnostic approval of pembrolizumab for patients with dMMR advanced tumor [33]. Moreover, several pieces of evidence showed no benefit or harm from peri-/post-operative chemotherapies in resectable MSI-H G/GEJ cancers.

In this regard, an exploratory subgroup analysis from the MAGIC trial showed that patients with an MSI-H/dMMR tumor had a better prognosis (mOS not reached 95% CI, 11.5-NR months) compared to those with an MSS tumor in the surgery-alone arm (mOS 20.5 months; 95% CI, 16.7–27.8 months; HR, 0.42; *p* = 0.09) [34]. Similarly, a post hoc analysis from the ITACA-S trial suggested that MSI-H status represents an independent prognostic factor, being associated with better DFS (*p* = 0.02) and OS (*p* = 0.01) [35]. It is noteworthy that a post hoc analysis from the CLASSIC trial showed that MSI-H (*p* = 0.008) and PD-L1 (*p* = 0.044) were independent prognostic factors and no DFS benefit was achieved by adjuvant chemotherapy for patients with an MSI-H tumor (5-year DFS 83.9% vs. 85.7%; *p* = 0.931) [36]. Finally, a meta-analysis focusing on patients with an MSI-H tumor including MAGIC, CLASSIC, ARTIST and ITACA-S trials was recently conducted. The authors described an advantage in DFS (5-year DFS 71.8% vs. 52.3%; *p* < 0.001) and OS (5-year OS 77.5% vs. 59.3%; *p* < 0.001) for patients with an MSI-H tumor compared to those with a microsatellite stable (MSS)/MSI-low tumor [37].

In addition, several prognostic factors have been suggested to sensitize to ICIs, such as EBV-positive, POLE/POLD1-mutated and high tumor mutational burden (TMB-H), but evidence is limited [38,39]. The results of the aforementioned trials provided the rationale for further investigations about ICI in the perioperative treatment of resectable G/GEJ cancer (Table 2).

#### Immunotherapy in Perioperative Treatment of G/GEJ Cancer

The turning point of adjuvant therapy in esophageal or GEJ cancer was represented by the phase III, global, randomized, double-blind, placebo-controlled arm CheckMate 577 trial. The study included patients with stage II/III esophageal or GEJ cancer. After chemoradiotherapy and radical surgery, patients with at least ypT1 and/or ypN1 histologically confirmed cancer were randomized to receive nivolumab 240 mg every 2 weeks for 16 weeks, then 480 mg every 4 weeks (up to 1 year of therapy) or placebo. The primary endpoint was DFS, which was found to be superior in the experimental arm for all the intention-to-treat (ITT) population (22.4 vs. 11.0 months; HR, 0.69; 95% CI, 0.56–0.86; *p* < 0.001). The benefit was observed in all subgroups, regardless of histological type and PD-L1 level. However, in the subgroup of GEJ patients, the benefit was less marked, with an mDFS of 22.4 months in the nivolumab arm vs. 20.6 months in the placebo arm (HR 0.87, IC 95% 0.63–1.21) as compared to that obtained in esophageal cancer (mDFS 24.0, vs. 8.3, respectively, HR 0.61) [40].

In the phase II PERFECT trial, 40 patients with resectable esophageal adenocarcinoma received atezolizumab in combination with chemoradiotherapy followed by surgery in 83% of cases. Immune-related AEs were observed in 15% (*n* = 6) of patients in the experimental arm. Ten patients achieved a partial response (PR). Despite being a single-arm study, this represented the first trial to propose the combination of the CROSS scheme and an anti-PD-L1 agent, and it proved the feasibility of adding atezolizumab to CROSS-based neoadjuvant chemoradiotherapy [41].

The efficacy of atezolizumab in the perioperative setting was also evaluated in the randomized multicenter phase IIb DANTE trial from the German and Swiss groups in which patients with resectable G/GEJ adenocarcinoma were assigned to receive atezolizumab in combination with FLOT or FLOT alone. The primary endpoint was progression-free survival (PFS)/DFS after 5 years of observation. Secondary endpoints were recently summarized in an interim analysis presented at the 2022 ASCO meeting, reporting atezolizumab plus FLOT as feasible and safe in the perioperative setting. Surgical mobility and mortality, and R0 rates were comparable between the two arms, while a higher rate of downsizing and pathological regression for the experimental arm (pT0, 23% vs. 15%; pN0, 68% vs. 54%) was reached, especially in patients whose tumor had a high PD-L1 expression or MSI-H status. Indeed, in the MSI-H and PD-L1 combined positive score (CPS) ≥ 10, tumor regression grade (TRG) 1a/b, assessed by central pathologists, was observed in 70–71% of cases in the experimental arm vs. 47–52% in the control arm. These data surely provide the rationale for implementation towards a phase III study [42,43].

The randomized, double-blind, phase III MATTERHORN trial aims at assessing the efficacy and safety of durvalumab in combination with perioperative FLOT, followed by adjuvant durvalumab monotherapy in patients with resectable G/GEJ cancer. The study is currently ongoing, with an estimated sample size of 900 patients at approximately 180 sites worldwide. The primary endpoint is event-free survival; secondary endpoints include OS and the pCR rate. Safety assessment will be evaluated [44].

A remarkable topic is the combination of immune checkpoint therapy, based on the possible synergistic effect of blocking both the PD-1/PD-L1 and the CTLA-4 pathways [45].

To this purpose, the phase II/III trial EA 2174, of the ECOG-ACRIN Cancer Research Group, is evaluating the perioperative treatment with nivolumab and ipilimumab in addition to standard chemotherapy and radiation therapy in patients with locoregional esophageal and GEJ adenocarcinoma. The primary neoadjuvant endpoint is the pCR rate, and the primary adjuvant endpoint is DFS. The safety run-in phase enrolled 31 patients divided into the two arms with no disproportional difference in safety or number of patients who proceeded to surgery [46].

The phase II VESTIGE trial has the primary objective to assess if the combination of nivolumab plus ipilimumab as an adjuvant treatment after neoadjuvant chemotherapy with FLOT and surgery improves DFS in patients with G/GEJ adenocarcinoma [47].

The immunotherapy combination strategy is also under evaluation in patients selected for biomarkers of response. The phase II NEONIPIGA trial enrolled 32 patients with resectable MSI-H/dMMR G/GEJ adenocarcinoma to receive neoadjuvant nivolumab in combination with ipilimumab for six cycles, followed by surgery and subsequent adjuvant treatment with single-agent nivolumab for 9 months. All patients received neoadjuvant treatment, and 29 of them proceeded to surgery. After a median follow-up of 12 months, 30 patients were free from progressive disease (PD), 1 patient died due to PD after five cycles of neoadjuvant therapy, and 1 patient died without relapse. Neoadjuvant therapy with dual checkpoint inhibition led to a pCR rate of 58.6%, meeting the primary endpoint. The combination of nivolumab and ipilimumab is promising as a neoadjuvant treatment in patients with MSI-H tumor and highlights the possibility of delaying or even avoiding surgery in highly selected subgroups of patients [48]. Furthermore, the Italian multicenter, non-randomized, single-arm, multi-cohort, open-label, phase II INFINITY study is evaluating the efficacy and tolerability of the combination of tremelimumab and durvalumab as a neoadjuvant (cohort 1) treatment or as a definitive treatment (cohort 2) in patients with MSI-H/dMMR resectable G/GEJ adenocarcinoma [49]. This study may provide new evidence on the therapeutic management of resectable G/GEJ cancer in patients selected for biomarkers highly predictive of response, particularly in regard to the possibility of non-surgical management.

Among the ongoing trials, the phase II, randomized IMAGINE trial (NCT04062656) is evaluating various immunotherapy or chemo-immunotherapy treatments (nivolumab, nivolumab plus ipilimumab, and nivolumab plus relatlimab, an anti-LAG3 antibody) in the perioperative set, while the KEYNOTE-585 trial (NCT03221426) is evaluating perioperative pembrolizumab plus chemotherapy for resectable G/GEJ adenocarcinoma. Notably, as there is no international standard adjuvant chemotherapy, pembrolizumab can be administered with cisplatin plus capecitabine or 5-fluorouracil, or, in a separate safety cohort, with the FLOT regimen.

The phase II, multicenter, 4-cohort, IMHOTEP trial (NCT04795661) is also evaluating the use of pembrolizumab in the neoadjuvant phase. Patients with resectable MSI-H/dMMR or EBV-positive GC will receive a single dose of pembrolizumab 400 mg 6 weeks before the surgery and a clinician’s choice adjuvant therapy. The primary endpoint will be pCR.

The efficacy of spartalizumab, an anti-PD-1 moAb, is under evaluation in association with perioperative chemotherapy with FLOT in the phase II ongoing GASPAR study [50] (Table 2).

### 2.3. Other Targeted-Therapies

GC is characterized by high intra- and inter-tumor heterogeneity, and the accumulation of new features is related to temporal and spatial progression, a condition that leads to the existence of numerous molecular variants. Precise molecular typing is a prerequisite for optimizing molecular targeted therapies [51]. Several targets are currently under investigation and though the prognostic role for some of them seems clear, few clinical trials have been designed in early settings.

Angiogenesis, particularly the vascular endothelial growth factor (VEGF) pathway, is a key player in tumor growth and metastatic spread in several neoplasms, including GC [52,53]. The VEGF family binds to vascular endothelial growth-factor receptors (VEGFR) (e.g., VEGFR1, VEGFR2 and VEGFR3). The VEGF/VEGFR pathway became an attractive target in GC and several drugs were tested in advanced disease, although unfortunately many of these agents did not demonstrate efficacy. Thus, bevacizumab, sorafenib and sunitinib have been studied demonstrating modest activity at the cost of known toxicity. Greater interest has raised VEGFR2 inhibition as a possible therapeutic target, and ramucirumab alone or in combination with taxane is currently a second-line standard treatment in patients with advanced GC [54,55]. In the neoadjuvant phase, apatinib (A), an anti-VEGFR2 compound, was investigated in addition to the FLOT (FLOTA) perioperative regimen in a Chinese retrospective dataset with the aim to estimate safety and efficacy. The results showed no significant differences in the primary endpoints (DCR, TRG and pCR) between FLOT and FLOTA probably due to the small sample size. However, the combination was associated with higher ORR compared to FLOT alone (80.6% vs. 50.0%, *p* = 0.008) without any specific alert in terms of safety [56]. Ongoing trials with A in combination with chemotherapy or other compounds are summarized in Table 3. Among these, a phase II exploratory trial (NCT03878472) investigated the activity of A in combination with camrelizumab and chemotherapy (S-1 ± oxaliplatin) in locally advanced GC. Although limited in sample size, the study showed that this drug combination was safe and provided pCR in 15.8% of cases and major pathological response in 26.3% [57].

Other compounds were previously investigated as per bevacizumab in the ST03 trial and ramucirumab in the RAMSES study; however, both of those studies failed to demonstrate benefits in their primary endpoint (DCR and ORR, respectively) [58,59]. Of note, ramucirumab was shown to improve R0-resection rates when added to perioperative FLOT but with a doubling rate of mortality (5.9% vs. 2.5%). Those discouraging results were not motivating in further pursuing the use of bevacizumab or ramucirumab in the perioperative setting. Fruquintinib is an orally highly selective small-molecule inhibitor of VEGFR1, VEGFR2 and VEGFR3. Its activity has been evaluated in gastrointestinal cancers and it is currently approved in China for treatment of chemorefractory metastatic colorectal cancer [60,61]. Waiting for the results of a phase III study of fruquintinib plus paclitaxel in second-line GC (FRUTIGA–NCT03223376), two phase II trials in a preoperative setting (Table 3) are already active and recruiting. Finally, vandetanib, an oral multikinase inhibitor directed against VEGFR2, epithelial growth factor receptor (EGFR) and RET was investigated in a phase I trial. The study aimed to determine the maximum tolerated dose (MTD) of vandetanib given concurrently with chemotherapy and radiation therapy followed by surgery. Nine patients with esophageal or GEJ cancer were enrolled and the study was able to determine vandetanib 100 mg/die as the recommended dose [62].

GCs frequently overexpress EGFR and despite its prognostic and predictive role being confirmed in several malignancies, the clinical and therapeutic implications in GC are still lacking. The EGFR signaling network consists of several overlapping and interconnecting hubs, including the phosphatidylinositol 3-kinase (PI3K)/AKT pathway, the RAS/RAF/MEK/ERK1/2 pathway, and the phospholipase C pathway. Each of these pathways plays an important role in mediating cell survival and proliferation, angiogenesis, adhesion and cell motility [63,64]. EGFR overexpression attests to 2–35% according to the literature and, although it might seem like an excellent candidate for combination therapies, it has been widely demonstrated that the use of specific drugs as per cetuximab and panitumumab did not provide a remarkable benefit in advanced GC [65,66,67,68]. Few experiences are available in the neoadjuvant setting with anti-EGFR. Cetuximab has been evaluated in a phase II single-arm multicenter trial, enrolling 65 patients. The addition of cetuximab to a neoadjuvant chemotherapy regimen was safe, but the study did not show sufficient efficacy to meet the primary endpoint [69]. Similar to the previous trial, the phase II study (NEOPECX) evaluating standard epirubicin, cisplatin and capecitabine (ECX) with or without panitumumab demonstrated a lack of benefit in terms of downstaging for locally advanced gastroesophageal cancer [70]. However, in this trial, as others cited before, the enrollment included GC and esophageal cancers that probably determine a confounding factor. Moreover, nimotuzumab, a humanized therapeutic monoclonal antibody against EGFR, demonstrated conflicting results in the neoadjuvant setting and a recently published meta-analysis concluded for a lack of benefit from this new compound [71]. According to a retrospective multi-institutional analysis, the possibility to adopt a hyperselection of patients with EGFR-amplified G/GEJ adenocarcinoma deserves special attention, resulting in exclusion of molecular alterations that could be responsible for intrinsic resistance [72].

Lenvatinib is a multikinase inhibitor including, among its targets, VEGFR1-3, fibroblast growth factor receptor (FGFR) 1–4, PDGFRα, and the oncogenes RET and KIT. Lenvatinib demonstrated promising activity in combination with the anti-PD-1 pembrolizumab, with a favorable safety profile, in previously treated advanced GC patients; thus, at ASCO 2022 a neoadjuvant phase II trial of lenvatinib plus pembrolizumab was presented [73,74].

Claudin18 isoform 2 (CLDN18.2) is a tight junction protein of the claudin (CLDN) family involved in molecular interchanges among cells. In gastric epithelium, CLDN18.2 has a barrier function preventing H+ leakage in the underlying cells layer [75]. The disruption of tight junctions during tumorigenesis leads to CLDN18.2 exposure on cells’ surface, making it vulnerable to targeting [76]. Although CLDN proteins are expressed on the surface of several neoplasms, CLDN18.2 expression is a prerogative of GC and is independent by HER2 amplification [77]. Zolbetuximab is a kimeric IgG moAb able to bind highly CLD18.2-expressed gastric cells inducing antibody-dependent cellular cytotoxicity (ADCC) and complement-dependent cytotoxicity apoptosis. The treatment demonstrates improvement of PFS and OS in the randomized phase II FAST trial when associated with epirubicin, oxaliplatin, capecitabine (EOX) in advanced CLDN18.2-positive GC [78]. Those results led to the planning of two phase-III trials testing the combination of zolbetuximab with oxaliplatin, 5-fluorouracil and leucovorin (mFOLFOX6) or capecitabine plus oxaliplatin (CAPOX) as first-line therapy (SPOTLIGHT-NCT03504397 and GLOW-NCT03653507). However, despite the encouraging results and promising efficacy of zolbetuximab, no trials in early setting are still available.

FGFR family members belong to the tyrosine kinase (TK) superfamily of membrane receptors. Each isoform is able to undergo dimerization after the bond with the fibroblast growth factor (FGF) ligand, leading to several downstream pathways’ activation as per MAPK/ERK and PI3K/AKT that are involved in cells’ proliferation and migration. GC experienced mainly FGFR2 alterations, followed by mutation on the other isoforms (FGFR1, 3 and 4). Amplification of FGFR2 is one of the principal mechanisms involved in GC carcinogenesis and it is related to poor prognosis [79]. Thus, the FGFR family has gained attention as a potential therapeutic anticancer target and countless early phase trials are exploring new compounds [80]. However, despite several attempts in clinical trials, only the phase III FIGHT trial showed improvement in both PFS and OS in HER2-negative, advanced GC when treated with FOLFOX and bemarituzumab at the cost of high ocular toxicity [81].

Another field of interest is the investigation of germline or somatic mutations in DNA damage repair (DDR) that led to tumorigenesis as the main mechanism of inactive or aberrant DNA repair. Historically, the most recognized DDR deficiencies are the germline or somatic BRCA1/2 mutations, representing the most corrupt in ovarian, pancreatic, prostatic and breast cancers. Despite the low rate of BRCA mutations in GC, other emerging proteins involved in DDR deficiency as per ataxia telangiectasia mutated (ATM) gene have been highlighted. The low rate of ATM has been identified in approximately 14% up to 22% of GC according to literature, and it is related to lymph nodes’ involvement, high-grade histology and poor 5-year outcome [82,83,84]. ATM localizes the double-stranded DNA breaks and with BRCA1 and p53-binding protein fixes up DNA damages, so low ATM promotes collection of DNA damages. In this regard, the use of platinum-based chemotherapy in these patients may be able to result in double-stranded DNA breaks that the tumor cell is unable to repair. Despite these prerogatives, data in clear favor of platinum-based treatments are contradictory in GC [85]. Preclinical models suggest an effect of poly(ADP-ribose) polymerase (PARP) inhibitors (PARPi) in GC cells, confirmed by the promising results of the randomized phase II trial in advanced GC designed to demonstrate the safety and efficacy of adding olaparib to taxane-based chemotherapy [82,86]. The failure of the phase III GOLD trial in advanced GC in the subgroup of ATM low patients did not lead to further development of PARPi in early stages and implied that in GC, the DDR mechanisms have a complexity that we do not yet fully understand [87].

Preclinical models showed a role in cancer suppression in GC cell lines of inhibition of TIM-3 and LAG-3 that might suggest a prognostic role of these factors [88]. A phase I study with the goal of evaluating tolerability of nivolumab and relatlimab (anti-LAG-3) is actually active, not recruiting (NCT03044613—Table 3) based on the retrospective observation that the overexpression of LAG-3 after nivolumab treatment correlates with improved outcome [89].

## 3. Discussion and Conclusions

The introduction of new drugs against potential molecular targets in advanced GC has paved the way to the concept of personalizing treatments in resectable disease. If the primary goals of treatment of advanced disease are to increase survival and maintain quality of life, the main objective of the perioperative approach is to increase the rate of disease-free patients. Furthermore, improving the outcome of patients amenable to perioperative strategy is an urgent need, as the disease recurrence rate with current treatments (e.g., CROSS or FLOT) is approximately 45–50% or even less outside the clinical trial framework [9,10,90].

Although personalized therapy appears to be a promising strategy, there are challenges ahead.

Firstly, given the rarity of several targets, study designs inevitably include small sample sizes. As a consequence, only large relative differences in primary outcome can be detected, while small differences or subgroup effects may be underestimated [16,91,92].

In studies involving molecularly selected subpopulations, another methodological question is the choice of the primary endpoints, a matter that becomes particularly relevant when it impacts on treatment decision-making. Given the limited number of patients potentially eligible for enrollment, some studies use tumor-centered endpoints, but where the intent of treatment is curative, the primary endpoints should be patient-centered [93]. In resectable gastric cancer, TRG has been considered to be closely associated with DFS and OS, and it has been proposed as a surrogate survival endpoint. Becker et al. analyzed 480 surgical specimens assessing histopathological tumor regression after neoadjuvant chemotherapy. Tumor regression has been identified as a parameter independently associated with survival outcome (*p* = 0.009), with regressive changes in lymph node metastases as strong predictor factors of OS [94,95]. Similarly, in the study of Lorenzen et al., the 3-year DFS was 71.8% in patients who experienced a pCR and 37.7% in those who had a residual disease on pathological specimen after neoadjuvant treatment and surgery (HR 0.38, *p* = 0.018) [96]. These findings were confirmed in a large Italian dataset (RealFLOT study) that prospectively collected clinical data from 206 patients treated with FLOT as perioperative chemotherapy. DFS and OS were significantly longer in patients who obtained TRG1a compared to other TRG (*p* = 0.009 and *p* = 0.023, respectively) [10]. Although being identified as a predictor of survival outcome, TRG remains a tumor-centered surrogate endpoint, while a survival gain, as primary endpoint, is required for practice-changing studies.

Overall, going through the evidence, two subsets of patients appear to be potential candidates for perioperative tailored treatments in the next future: HER2-positive and MSI-H.

The development of a perioperative treatment strategy directed against HER2 has been tortuous and, to date, inconclusive. After the disappointing results of the JACOB, TyTAN, TRIO-013/LOGiC and GATSBY studies, novel HER2-directed antibodies (e.g., T-DXd), tyrosine kinase inhibitors (e.g., tucatinib) or HER-Vaxx have appeared in the treatment scenario of metastatic GC, providing new opportunities to improve the clinical outcome of patients with a HER2-positive tumor [25,29,97,98,99,100]. A better understanding of the primary and secondary resistance mechanisms to anti-HER2 agents could facilitate the selection of patients most likely to benefit from them. Although the exact mechanisms are not yet fully understood, a recent integrated analysis of public datasets identified five hub genes (GNGT1, KRT7, KRT16, SOX9, TIMP1) implicated in trastuzumab-resistant pathways with a potentially relevant role as therapeutic biomarkers for GC [101].

Several HER2-directed agents are currently under evaluation as neoadjuvant treatment in resectable G/GEJ cancer, and their role in this clinical context will be further elucidated. While awaiting the results from the ongoing trials (e.g., EORTC 1203 INNOVATION, EPOC2003), HER2-directed regimens should not be recommended in routine clinical practice.

Growing evidence revealed that patients with resectable MSI-H GC had better prognosis and limited impact from perioperative chemotherapy [34,35,36,37]. Although the role of the perioperative taxane-based triplet in this subgroup of patients remains to be defined, the results from ICI-based therapy in metastatic disease provide a strong rationale to offer immunotherapeutic agents also in the neoadjuvant phase [10,14,102]. The INFINITY trial is investigating the role of a chemo-free neoadjuvant therapy in patients with resectable MSI-H G/GEJ adenocarcinoma. The primary endpoint for cohort 2 is a 2-year complete response rate (CRR), defined as the absence of macroscopic or microscopic residual disease (locally, regionally and distantly) at radiological examinations, tissue and liquid biopsy, in the absence of salvage gastrectomy. This study introduces the concept of non-operative management for patients with a complete clinical–pathological–molecular response, and this is a novel and potential practice-changing strategy [49]. The high reliability of MSI-H as a biomarker of response to ICI makes MSI-H GC patients the best candidates for the paradigm shift from chemotherapy plus surgery to immunotherapy as a definitive treatment. However, some questions deserve special attention, such as whether the chemo-free strategy is applicable for any extent of disease including patients with poor prognosis (T4 and/or N2-3); in addition, efforts should be made to understand the resistance mechanisms in patients who receive a suboptimal benefit from treatment.

As previously described, CLDN18.2 is a promising target for drug development [103]. The first-in-class moAb zolbetuximab is currently under investigation in patients with CLDN18.2-positive, HER2-negative, advanced G/GEJ cancer. If the results of the SPOTLIGHT (NCT03504397) and GLOW trials (NCT03653507) confirm the expectations, moAbs directed against CLDN18.2 could be integrated into clinical trials in the early setting, as well as changing the clinical practice for patients with metastatic disease.

As suggested by the PANGEA study, a better optimization of gastric cancer treatment allows maximizing patient outcomes [51]. Recent advances in metastatic disease led to a strong recommendation for the addition of nivolumab in combination with first-line chemotherapy in patients with HER2-negative GC with PD-L1 CPS ≥ 5. However, the choice of the cutoff was based on subgroup analyses and the optimal value remains unclear. Other ASCO recommendations include pembrolizumab plus chemotherapy as first-line treatment in patients with HER2-negative esophageal/GEJ adenocarcinoma with PD-L1 CPS ≥ 10, the combination of trastuzumab, pembrolizumab and chemotherapy as first-line treatment in patients with HER2-positive G/GEJ adenocarcinoma, and T-DXd as second- or further-line treatment in patients with HER2-positive G/GEJ adenocarcinoma [104]. As revealed by multi-omics studies and clinical practice data, the primary tumor location is associated with a heterogeneous molecular landscape and different clinical behavior. Beyond the clinical issues, the site of the primary tumor can guide towards a treatment strategy also on the basis of the likely presence of biomarkers of response. In a recent multi-omics study, patients with specific proteomic-based subtypes had a different expression of immune checkpoint genes and may have a different response to ICIs. Further studies are needed to fully understand the implications of the primary tumor location and respective proteomic profile on the therapeutic decision [105].

In the perioperative setting, although caution is mandatory, times are changing and tailored strategies could introduce new treatment concepts. To date, MSI-H GC patients possess the characteristics to be the subgroup that could receive the best benefit from a tailored approach.

## Figures and Tables

**Figure 1 ijms-24-04877-f001:**
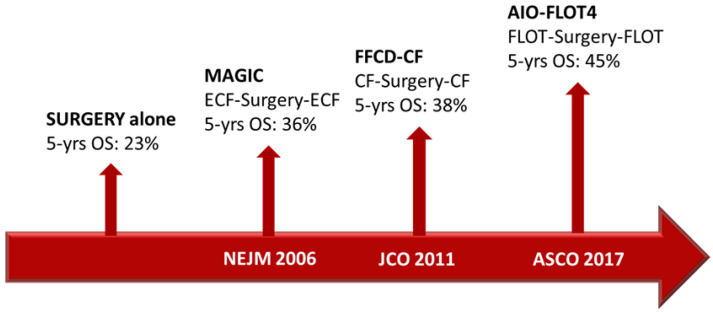
Timeline of the main trials that change the clinical practice in perioperative GC treatment.

**Table 1 ijms-24-04877-t001:** Summary of the main trials with anti-HER2 agents in resectable G/GEJ cancer.

Trial	Phase	NCT	Treatment Setting	Arms	Primary End-Point	Status
PETRARCA	II	NCT02581462	Perioperative	FLOT alone FLOT, trastuzumab, pertuzumab	pCR	Completed
EPOC2003	II	NCT05034887	Neoadjuvant	T-DXd	Major pathological response rate	Recruiting
EORTC 1203 INNOVATION	II	NCT02205047	Perioperative	Cisplatin, capecitabine/5-fluoruracil Cisplatin, capecitabine/5-fluoruracil, trastuzumab Cisplatin, capecitabine/5-fluoruracil, trastuzumab, pertuzumab	Major pathological response rate	Active, not recruiting
Trigger Study	II	UMIN 000016920.	Perioperative	S-1, cisplatin S-1, cisplatin, trastuzumab	OS	Completed
HERFLOT	II	NCT01472029	Perioperative	FLOT trastuzumab FLOT alone	pCR	Completed
RG Oncology/RTOG 1010	III	NCT01196390	Neoadjuvant	Carboplatin, paclitaxel, trastuzumab plus radiotherapy Carboplatin, paclitaxel plus radiotherapy	DFS	Active, not recruiting

Abbreviations: FLOT: fluorouracil, leucovorin, oxaliplatin and docetaxel; G/GEJ: gastric or gastro-esophageal junction; HER2: human epidermal growth factor receptor-2; OS: overall survival; pCR: pathological complete response; T-DXd trastuzumab deruxtecan.

**Table 2 ijms-24-04877-t002:** Summary of the main trials with ICI in resectable G/GEJ cancer.

Trial	Phase	Treatment Setting	Arms	Primary End-Point	Status
Checkmate 577	III	Adjuvant	Nivolumab Placebo	DFS	Active, not recruiting
DANTE	II	Perioperative	Atezolizumab, FLOT FLOT	DFS/PFS	Active, not recruiting
MATTERHORNE	III	Perioperative	FLOT, durvalumab FLOT	EFS	Recruiting
NEONIPIGA	II	Perioperative	Nivolumab, ipilimumab	pCR	Recruiting
EA 2174	II/III	Perioperative	Carboplatin, paclitaxel, radiation therapy Carboplatin, paclitaxel, radiation therapy, nivolumab Nivolumab Nivolumab-Ipilimumab	pCR	Recruiting
VESTIGE	II	Adjuvant	Chemotherapy according to guidelines Nivolumab + ipilimumab	DFS	Active, non recruiting
INFINITY	II	Neoadjuvant (Cohort 1) and definitive treatment (Cohort 2)	Tremelimumab, durvalumab	Cohort 1: pCR and negative ctDNA status Cohort 2: 2-year CRR	Recruiting
IMAGINE	II	Perioperative	Nivolumab, FLOT Nivolumab, relatlimab	pCR	Recruiting
Keynote-585	III	Perioperative	Pembrolizumab, chemotherapy Placebo, chemotherapy	EFS, pCR, OS, AEs	Active, non recruiting
IMHOTEP	II	Neoadjuvant	Pembrolizumab	pCR	Recruiting
GASPAR	II	Perioperative	FLOT, spartalizumab	pCR	Recruiting

AE: adverse event; CRR: complete response rate; DCR: disease control rate; DFS: disease-free survival; dMMR: deficient mismatch repair; EFS: event-free survival; FLOT: fluorouracil, leucovorin, oxaliplatin and docetaxel; G/GEJ: gastric or gastro-esophageal junction; ICI: immune checkpoint inhibitor; OS: overall survival; pCR: pathological complete regression; PFS: progression-free survival.

**Table 3 ijms-24-04877-t003:** Summary of the main trials on agents directed against novel targets in resectable G/GEJ cancer.

Phase	NCT	Regimen	Recruitment Status	Country
Phase II	NCT03878472	Camrelizumab ± apatinib ± SOX	Recruiting	China
Phase II	NCT03229096	Apatinib + CAPOX	Unknown	China
Phase II	NCT03192735	Apatinib + SOX	Active, not recruiting	China
Phase II	NCT05223088	Tislelizumab + apatinib + SOX	Recruiting	China
Phase II	NCT04195828	Camrelizumab + apatinib + nab-paclitaxel and S-1	Recruiting	China
Phase II	NCT05122091	Fruquintinib + SOX	Recruiting	China
Phase II	NCT05177068	Fruquintinib + sintilimab + SOX	Active, not yet recruiting	China
Phase I	NCT03044613	Nivolumab or nivolumab/relatlimab	Active, not yet recruiting	USA

Abbreviations: CAPOX: capecitabine plus oxaliplatin; G/GEJ: gastric or gastro-esophageal junction; SOX: S-1 and oxaliplatin.

## Data Availability

All data generated or analyzed during this study are included in this published article.

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
