# Peer review of "Perioperative Tailored Treatments for Gastric Cancer: Times Are Changing"

_ijms, 2023, doi:10.3390/ijms24054877_

Round 1

Reviewer 1 Report

Interesting review. I recommend authors to include Table 2 in the paragraph "Immunotherapy in the perioperative treatment of G/GEJ cancer" which should be 2.3 (accordingly, the numbers of the following paragraphs should change) since it summarizes the main studies related to immunotherapy.

Also, I suggest adding a new table in section 2.2 summarizing the main studies with ICI in resectable G/GEJ cancer. 

I also suggest minimal errors and recommend checking spaces and full stop especially near references.

Additionally, authors should provide the missing information in the following sections.

Author Contributions:

Financing:

Institutional Review Board Statement:

Declaration of informed consent:

Data Availability Statement:

Thank you:

Conflict of Interest:

Author Response

Interesting review. I recommend authors to include Table 2 in the paragraph "Immunotherapy in the perioperative treatment of G/GEJ cancer" which should be 2.3 (accordingly, the numbers of the following paragraphs should change) since it summarizes the main studies related to immunotherapy.

Also, I suggest adding a new table in section 2.2 summarizing the main studies with ICI in resectable G/GEJ cancer.

We included Table 2 also in the paragraph “"Immunotherapy in the perioperative treatment of G/GEJ cancer" and we named this paragraph “2.2.1”, since the main trials with ICI in resectable G/GEJ cancer are included in this table (ongoing status).

I also suggest minimal errors and recommend checking spaces and full stop especially near references.

We check our paper for minimal error correction.

Additionally, authors should provide the missing information in the following sections.

Author Contributions:

Financing:

Institutional Review Board Statement:

Declaration of informed consent:

Data Availability Statement:

Thank you

Conflict of Interest

We added the missing information as follows

Author Contributions:

All authors were involved in conceptualizing the manuscript; the introduction was compiled by AG, “HER2-directed therapies” by GV, “Immune checkpoint blockade” by EB, “Other target-ed-therapies” by SF, and Discussion and conclusions by DL. LA, DL, SF, GR, CW, EC, AV, EG, MB, FC, and SP revised all the manuscript. All authors also approved submission of the manuscript. All authors have read and agreed to the published version of the manuscript.

Funding:

No financial funding was received.

Institutional Review Board Statement:

Not applicable.

Informed Consent Statement:

Not applicable.

Data Availability Statement:

All data generated or analyzed during this study are included in this published article.

Acknowledgments:

Not applicable.

Conflicts of Interest:

The authors declare that they have no competing interests.

Reviewer 2 Report

In this article, Lavacchi and colleagues aim to provide a review of the state of the art concerning potential tailored therapeutics in gastric cancer.

·      Overall, I find this article to be very relevant and comprehensive, covering the major advances in gastric cancer treatment strategies and trials.

·   However, I found the article quite descriptive and confusing, lacking critical discussion of the trials presented. More so, the concluding messages were somewhat contradictory, namely:

“Overall, summarizing the evidence, two subsets of patients appear to be potentially candidates for perioperative tailored treatments in the next future: HER2-positive and MSI-H.”

“In the perioperative setting, although caution is mandatory, times are changing and tailored strategies could introduce new treatment concepts. To date, MSI-H GC patients possess the characteristics to be the sub-group that could receive the best benefit from a tailored approach.”

   ·   The abstract in particular should be re-written and more focused.

Author Response

In this article, Lavacchi and colleagues aim to provide a review of the state of the art concerning potential tailored therapeutics in gastric cancer.

  •     Overall, I find this article to be very relevant and comprehensive, covering the major advances in gastric cancer treatment strategies and trials.
  •  However, I found the article quite descriptive and confusing, lacking critical discussion of the trials presented. More so, the concluding messages were somewhat contradictory, namely:

“Overall, summarizing the evidence, two subsets of patients appear to be potentially candidates for perioperative tailored treatments in the next future: HER2-positive and MSI-H.”

“In the perioperative setting, although caution is mandatory, times are changing and tailored strategies could introduce new treatment concepts. To date, MSI-H GC patients possess the characteristics to be the sub-group that could receive the best benefit from a tailored approach.”

  • The abstract in particular should be re-written and more focused.

MSI-H GC patients could receive the best benefit from perioperative ICI-based treatment. HER2-positive GC patients could be potentially candidates but conflicting results require caution. We expressed this concept with this sencence “While awaiting the results from the ongoing trials (e.g. EORTC 1203 INNOVATION, EPOC2003), HER2-directed regimens should not be recommended in routine clinical practice.”

We re-write the abstract adding the concept of non-operative management, as follows:

Resectable gastric or gastroesophageal (G/GEJ) cancer is a heterogeneous disease with no defined molecularly-based treatment strategy. Unfortunately, nearly half of patients experience disease recurrence despite standard treatments (neoadjuvant and/or adjuvant chemothera-py/chemoradiotherapy and surgery). In this review, we summarize the evidence of potential tailored approaches in perioperative treatment of G/GEJ cancer, with a special focus on patients with human epidermal growth factor receptor-2(HER2)-positive and microsatellite instabil-ity-high (MSI-H) tumors. In patients with resectable MSI-H G/GEJ adenocarcinoma, the ongoing INFINITY trial introduces the concept of non-operative management for patients with complete clinical-pathological-molecular response and this could be a novel and potential practice changing strategy. Other pathways involving vascular endothelial growth factor receptor (VEGFR), fibro-blast growth factor receptor (FGFR), claudin18 isoform 2 (CLDN18.2), and DNA damage repair proteins are also described, with limited evidence until now. Although tailored therapy appears to be a promising strategy for resectable G/GEJ cancer, there are several methodological issues to address: inadequate sample size for pivotal trials, underestimation of subgroup effects, and choice of primary endpoint (tumor-centered vs. patient-centered endpoints). A better optimization of G/GEJ cancer treatment allows to maximize patient outcomes. In the perioperative phase, although caution is mandatory, times are changing and tailored strategies could introduce new treatment concepts. Overall, MSI-H G/GEJ cancer patients possess the characteristics to be the subgroup that could receive the most benefit from a tailored approach.

Reviewer 3 Report

In view of recent advances in the field of gastric cancer treatment, this review is welcome. However the authors are facing a considerable challenge and  there is still substantial room for improvement . Revisions are needed to clarify the text, update and incorporate important references, prioritise and harmonize important concepts and recommendations. Remarks, comments, questions are as follows:

1- Introduction: In order to get an evolutionary representation of the practices, and to prioritise the relevant literature, a timeline representation is needed that would  facilitate reading and serve as a guideline for the reader. 

2 - several sentences are unintelligible. For example, what about « Subsequently, Kurokawa and colleagues demonstrated that HER2 overexpression, regardless of the immunohistochemistry (IHC)… » / « patients (25.0%) had subtotal response (<10% vital tumor cells) » (vital or viable?) / 

2 - Her-2: The authors should briefly discuss trastuzumab resistance (see for example Chen et al. Gene 2023). What about the clinical development (if any) of HER-vaxx? 

3 - Neoadjuvant therapy: the significance of histopathological tumor regression is cited only in a 2011 paper (ref 94). The authors should update  and improve this topic ( among other examples see Langer and Becker Virchows Arch. 2018; Reim et al. L. Pathol Clin Res 2020) 

The authors should comment the exploratory trial NCT03878472  (see Table 3) aimed at evaluating the efficacy of a combination of immune checkpoint immunotherapy antiangiogenesis and chemotherapy (citation: Li S et al. Nat Commun 2023). 

4 - The authors should cite and comment the recently issued ASCO guideline regarding immunotherapy and targeted therapy for advanced gastroesophagal cancer (see Shah et al. J. Clin. Oncol. 2023) 

5 - The concept of non-operative management for patients with complete clinical-pathological-molecular responses could be illustrated by  Peng et al . Patients with positive HER-2 amplification advanced gastroesophageal junction cancer achieved complete response with combined chemotherapy of AK104/cadonilimab (PD-1/CTLA-4 bispecific): A case report ». Needs a comment.

5 - The authors should introduce and discuss the concept of adenocarcinoma of esophagogastric junction as a distinct entity based on specific vulnerabilities. See the recent article By Li et al. Integrative proteomic characterization of adenocarcinoma of esophagogastric junction. Nat Commun 2023)

6 - About patients with EGFR-amplified gastric adenocarcinoma:  Maron et al. Epidermal Growth Factor Receptor Inhibition in Epidermal Growth Factor Receptor-Amplified Gastroesophageal Cancer: Retrospective Global Experience. J. Clin Oncol. 2022

Author Response

In view of recent advances in the field of gastric cancer treatment, this review is welcome. However the authors are facing a considerable challenge and  there is still substantial room for improvement . Revisions are needed to clarify the text, update and incorporate important references, prioritise and harmonize important concepts and recommendations. Remarks, comments, questions are as follows:

1- Introduction: In order to get an evolutionary representation of the practices, and to prioritise the relevant literature, a timeline representation is needed that would  facilitate reading and serve as a guideline for the reader. 

As suggested, we added Fig1 reporting a timeline of the main trials which led to the major change in clinical practice. 

2 - several sentences are unintelligible. For example, what about « Subsequently, Kurokawa and colleagues demonstrated that HER2 overexpression, regardless of the immunohistochemistry (IHC)… » / « patients (25.0%) had subtotal response (<10% vital tumor cells) » (vital or viable?) / 

We reviewed our paper with a native English-speaking colleague making several substantial changes.

We modified these sentences as follows:

“Subsequently, Kurokawa and colleagues demonstrated that HER2 overexpression was an independent prognostic factor in patients with resectable GC (HR 1.59; 95% CI 1.24 - 2.02; p=0.001) and they also showed that HER2 intra-tumoral heterogeneity was frequent and did not affect prognosis.”

“This trial reached the primary endpoint (pCR >20%) achieving a pCR rate of 21.4%, in fact 12 out of 56 patients had no viable neoplastic cells at the time of surgery, as assessed by central pathologist, and 14 patients (25.0%) had subtotal response (<10% residual tumor).”

2 - Her-2: The authors should briefly discuss trastuzumab resistance (see for example Chen et al. Gene 2023). What about the clinical development (if any) of HER-vaxx? 

As suggested we added the following sentences in the discussion:

The development of a perioperative treatment strategy directed against HER2 has been tortuous and, to date, inconclusive. After the disappointing results of the JACOB, TyTAN, TRIO-013/LOGiC and GATSBY studies, novel HER2-directed antibodies (e.g. T-DXd), tyrosine kinase inhibitors (e.g. tucatinib) or HER-Vaxx have appeared in the treatment scenario of metastatic GC giving a new opportunity to improve the clinical out-come of patients with HER2-positive tumor. [26, 30, 96 - 99]. A better understanding of the primary and secondary resistance mechanisms to anti-HER2 agents could facilitate the selection of patients most likely to benefit from them. Although the exact mechanisms are not yet fully understood, a recent integrated analysis of public datasets identified five hub genes (GNGT1, KRT7, KRT16, SOX9, TIMP1) implicated in trastuzumab-resistant path-ways with a potentially relevant role as therapeutic biomarkers for GC. [Chen F, Wang Y, Zhang X, et al. Five hub genes contributing to the oncogenesis and trastuzumab-resistance in gastric cancer. Gene. 2023 Jan 30;851:146942. doi: 10.1016/j.gene.2022.146942. Epub 2022 Oct 3. PMID: 36202277.]

3 - Neoadjuvant therapy: the significance of histopathological tumor regression is cited only in a 2011 paper (ref 94). The authors should update  and improve this topic ( among other examples see Langer and Becker Virchows Arch. 2018; Reim et al. L. Pathol Clin Res 2020) 

The authors should comment the exploratory trial NCT03878472  (see Table 3) aimed at evaluating the efficacy of a combination of immune checkpoint immunotherapy antiangiogenesis and chemotherapy (citation: Li S et al. Nat Commun 2023).

We integrated in the discussion about the TRG, the recent study of Reim and collegues.

“In resectable gastric cancer, TRG has been considered to be closely associated with DFS and OS and it has been proposed as a surrogate survival endpoint. Becker et al analyzed 480 surgical specimens assessing histopathological tumor regression after neoadjuvant chemotherapy. Tumor regression has been identified as a parameter independently asso-ciated with survival outcome (p=0.009) with regressive changes in lymph node metastases as a strong predictor of OS”

“Reim D, Novotny A, Friess H, et al. Significance of tumour regression in lymph node metastases of gastric and gas-tro-oesophageal junction adenocarcinomas. J Pathol Clin Res. 2020 Oct;6(4):263-272. doi: 10.1002/cjp2.169. Epub 2020 May 13. PMID: 32401432; PMCID: PMC7578278. “

We added the recent paper on MSI-H patients treated with FLOT in a real-life setting

“Nappo F, Fornaro L, Pompella L, et al. Pattern of recurrence and overall survival in esophagogastric cancer after perioperative FLOT and clinical outcomes in MSI-H population: the PROSECCO Study. J Cancer Res Clin Oncol. 2023 Feb 16. doi: 10.1007/s00432-023-04636-y. Epub ahead of print. PMID: 36795195.”

We also added the following sentences on NCT03878472 trial

“Li S, Yu W, Xie F, et al. Neoadjuvant therapy with immune checkpoint blockade, antiangiogenesis, and chemotherapy for locally advanced gastric cancer. Nat Commun. 2023 Jan 3;14(1):8. doi: 10.1038/s41467-022-35431-x. PMID: 36596787; PMCID: PMC9810618.”

4 - The authors should cite and comment the recently issued ASCO guideline regarding immunotherapy and targeted therapy for advanced gastroesophagal cancer (see Shah et al. J. Clin. Oncol. 2023) 

We reported the recent advances in immunotherapy and targeted therapy for advanced G/GEJ cancer, as suggested by the reviewer.

Recent advances in metastatic disease led to a strong recommendation for the addition of nivolumab in combination with first-line chemotherapy in patients with HER2-negative GC with PD-L1 CPS ≥ 5. However, the choice of the cutoff was based on subgroup anal-yses and the optimal value remains unclear. Other ASCO recommendations include pembrolizumab plus chemotherapy as first-line treatment in patients with HER2-negative esophageal/GEJ adenocarcinoma with PD-L1 CPS ≥ 10, the combination of trastuzumab, pembrolizumab and chemotherapy as first-line treatment in patients with HER2-positive G/GEJ adenocarcinoma, and T-DXd as second- or further-line treatment in patients with HER2-positive G/GEJ adenocarcinoma. [Shah MA, Kennedy EB, Alarcon-Rozas AE, et al. Immunotherapy and Targeted Therapy for Advanced Gastroesophageal Cancer: ASCO Guideline. J Clin Oncol. 2023 Jan 5:JCO2202331. doi: 10.1200/JCO.22.02331. Epub ahead of print. PMID: 36603169.]

5 - The concept of non-operative management for patients with complete clinical-pathological-molecular responses could be illustrated by  Peng et al . Patients with positive HER-2 amplification advanced gastroesophageal junction cancer achieved complete response with combined chemotherapy of AK104/cadonilimab (PD-1/CTLA-4 bispecific): A case report ». Needs a comment.

We added a sentence on the combination of trastuzumab, pembrolizumab and chemotherapy as first-line treatment in patients with HER2-positive G/GEJ adenocarcinoma (Shah et al 2023), but both the KN811 and the case report of Peng et al regard advanced disease. The non-operative management in this paper is a proposal for patients with resectable disease.

5 - The authors should introduce and discuss the concept of adenocarcinoma of esophagogastric junction as a distinct entity based on specific vulnerabilities. See the recent article By Li et al. Integrative proteomic characterization of adenocarcinoma of esophagogastric junction. Nat Commun 2023)

As revealed by multi-omics studies and clinical practice data, the primary tumor location is associated with a heterogeneous molecular landscape and different clinical behavior. Beyond the clinical issues, the site of the primary tumor can guide towards a treatment strategy also on the basis of the likely presence of biomarkers of response. In a recent multi-omics study, patients with specific proteomic-based subtypes had a different expression of immune checkpoint genes and may have a different response to ICIs. Further studies are needed to fully understand the implications of the primary tumor location and respective proteomic profile on the therapeutic decision. [Li S, Yuan L, Xu ZY, et al. Integrative proteomic characterization of adenocarcinoma of esophagogastric junction. Nat Commun. 2023 Feb 11;14(1):778. doi: 10.1038/s41467-023-36462-8. PMID: 36774361; PMCID: PMC9922290.]

6 - About patients with EGFR-amplified gastric adenocarcinoma:  Maron et al. Epidermal Growth Factor Receptor Inhibition in Epidermal Growth Factor Receptor-Amplified Gastroesophageal Cancer: Retrospective Global Experience. J. Clin Oncol. 2022

We added the following sentences:

“According to a restrospective multi-institutional analysis, deserve a special attention the possibility to adopt an hyperselection of patients with EGFR-amplified G/GEJ adenocar-cinoma, excluding molecular alterations that could be responsible for intrinsic resistance.” [Maron SB, Moya S, Morano F, et al. Epidermal Growth Factor Receptor Inhibition in Epidermal Growth Factor Receptor-Amplified Gastroesophageal Cancer: Retrospective Global Experience. J Clin Oncol. 2022 Aug 1;40(22):2458-2467. doi: 10.1200/JCO.21.02453. Epub 2022 Mar 29. PMID: 35349370; PMCID: PMC9467681.]

Round 2

Reviewer 3 Report

Clearly this new version of the MS contains improvements.